# Cryptic genetic variation in a heat shock protein modifies the outcome of a mutation affecting epidermal stem cell development in *C. elegans*

Sneha L. Koneru[1], Mark Hintze[1,2], Dimitris Katsanos[1,2] & Michalis Barkoulas [1✉]

A fundamental question in medical genetics is how the genetic background modifies the phenotypic outcome of mutations. We address this question by focusing on the seam cells, which display stem cell properties in the epidermis of *Caenorhabditis elegans*. We demonstrate that a putative null mutation in the GATA transcription factor *egl-18*, which is involved in seam cell fate maintenance, is more tolerated in the CB4856 isolate from Hawaii than the lab reference strain N2 from Bristol. We identify multiple quantitative trait loci (QTLs) underlying the difference in phenotype expressivity between the two isolates. These QTLs reveal cryptic genetic variation that reinforces seam cell fate through potentiating Wnt signalling. Within one QTL region, a single amino acid deletion in the heat shock protein HSP-110 in CB4856 is sufficient to modify Wnt signalling and seam cell development, highlighting that natural variation in conserved heat shock proteins can shape phenotype expressivity.

[1] Department of Life Sciences, Imperial College, London, United Kingdom. [2]These authors contributed equally: Mark Hintze, Dimitris Katsanos. ✉email: m.barkoulas@imperial.ac.uk

Since the completion of the human genome project, there has been a long-standing interest in using genomic data to predict phenotypes. Understanding the genotype-to-phenotype relationship represents one of the goals of personalised medicine, aimed at exploiting the predictability in this relationship to determine an individual's predisposition to disease or response to therapeutic treatment. Nevertheless, a fundamental challenge in this endeavour stems from the fact that genes exert their effects via complex gene-by-gene interactions[1], therefore genetic variation present in the background can modify the genotype-to-phenotype mapping. Even when considering monogenic disorders, genetic modifiers can modulate the penetrance or expressivity of disease phenotypes, thereby influencing the proportion of the population developing the disease (incomplete penetrance) or the severity of the disease for each individual in the population (variable expressivity[2]).

With regard to model organisms, mutant alleles are commonly studied in a single reference strain. Lab reference strains are highly inbred and have been selected to minimise genetic variation, which can act as a confounder in the interpretation of genetic results. However, laboratory-driven evolution can fix alleles that are not broadly present in the wild[3,4]. In *Caenorhabditis elegans*, the laboratory reference strain N2 (isolated from Bristol, UK), has several adaptations suited to the lab environment[5]. For example, N2 carries laboratory-derived alleles for genes such as *npr-1*, which influences a large number of phenotypes and *nath-10*, which affects life-history traits[5,6]. As a consequence, it is possible that well-studied mutations in N2 could have different phenotypic outcomes if they were to be studied in different genetic backgrounds. Such differences can vary from subtle phenotypic effects to substantial changes, for example, some genes considered essential in one *S. cerevisiae* isolate can be dispensable for survival in another[7]. Investigating how the effects of genetic mutations may vary across genetic backgrounds is important to uncover new gene functions and understand how genetic modifiers can shape phenotypic traits.

We study this problem using seam cell development in *C. elegans* as a simplified model of stem cell patterning[8]. The seam cells are lateral epidermal stem cells that contribute to the production of the syncytial hypodermis, as well as produce neuronal lineages. Seam cell development is fairly invariant in the lab reference N2 with stereotypic symmetric and asymmetric divisions during post-embryonic development giving rise to 16 cells per lateral side (Fig. 1a)[9,10]. Over recent years, *C. elegans* has been sampled from around the globe[11], which offers exciting opportunities to study how the genetic background affects development. We have recently shown that seam cell development is robust to standing genetic variation, with genetically divergent *C. elegans* isolates displaying a comparable seam cell number (SCN) to the lab reference strain[12]. Nevertheless, developmental robustness can lead to accumulation of genetic variation, which may not manifest phenotypically in normal conditions, but can be revealed upon perturbation[13,14]. This is called cryptic genetic variation and is a hidden source of variation that can influence many phenotypes and complex disease[13,15,16].

Mutations in genes affecting seam cell development have been studied so far only in the N2 background, with a number of transcription factors shown to play a role in this model[8]. Among them, GATA-type transcription factors are evolutionary conserved regulators of proliferation, and mutations in these factors have been linked to disease, such as aggressive breast, colorectal, and lung cancers[17]. In *C. elegans*, GATA transcription factors play crucial roles in the development of the gut, epidermis and vulva. One particular pair, EGL-18 and its closely related paralogue ELT-6, are thought to specify seam cell fate and are direct targets of the Wnt signalling pathway through the effector POP-1/TCF[18]. Mutations in *egl-18* result in precocious seam cell differentiation, which has been shown in embryos to correlate with misexpression of hypodermal markers in seam cells[19]. Furthermore, EGL-18 is required for the seam cell hyperplasia observed upon Wnt signalling hyperactivation, for example, following *pop-1* RNA interference (RNAi), which results in symmetrisation of normally asymmetric cell divisions expanding the SCN[18,20].

We sought to investigate how the genetic background influences the phenotypic outcome of mutations disrupting seam cell development. We demonstrate here that a loss-of-function mutation in the transcription factor *egl-18* results in higher average SCN in the CB4856 strain from Hawaii compared with N2. Using quantitative genetics, we map the genetic basis of the difference in the outcome of the mutation. We identify a complex genetic architecture with multiple quantitative trait loci (QTLs) affecting phenotype expressivity in the presence of the mutation, whereas being cryptic for seam cell development in a wild-type background. We show that these QTLs potentiate the effect of the Wnt signalling pathway in seam cell fate maintenance. We finally show that in CB4856 a single amino-acid deletion in HSP-110, which is a heat shock 70 family member, contributes to the difference in phenotype expressivity between the two isolates. Our results highlight that natural variation in a conserved heat shock protein is a key determinant of the outcome of a mutation influencing stem cell behaviour by potentiating a conserved signalling pathway.

## Results

**A mutation in the GATA transcription factor *egl-18* leads to a milder decrease in SCN in CB4856 than N2.** To investigate whether the genetic background can influence the phenotypic outcome of mutations affecting seam cell development, we introduced genetic perturbations targeting seam cell regulators, known from previous research in N2, into two wild isolates of *C. elegans*. We chose the commonly used divergent isolate CB4856 from Hawaii and isolate JU2007, which we had previously sampled from the Isle of Wight in the UK. Over the course of these experiments, we discovered that a putative null mutation in the GATA transcription factor *egl-18* (*ga97*, R193STOP), which acts downstream of the Wnt signalling pathway to maintain the seam cell fate[18,19], led to a milder decrease in SCN in the CB4856 background compared to N2 (Fig. 1b). Based on the distributions, there was higher phenotype expressivity in N2, however, more *egl-18(ga97)* animals showed the wild-type SCN in the CB4856 background than N2, thus we cannot rule out a concurrent change in phenotypic penetrance (Fig. 1b). Importantly, the difference between the two isolates was very reproducible in independent experiments with approximately two extra seam cells produced on average in the CB4856 background carrying the *egl-18(ga97)* mutation than N2 (Supplementary Fig. 1a). Interestingly, there was no significant difference in phenotype expressivity between N2 and JU2007 with CB4856 being distinct among these strains, which suggested that the underlying cause might not be specific to N2 and its lab domestication (Supplementary Fig. 1b).

The difference between N2 and CB4856 was more apparent upon strong loss-of *egl-18* function, as it was less pronounced upon introgression of the milder *egl-18(ok290)* allele that represents an in-frame 698 bp deletion (Supplementary Fig. 1b). Perturbing other components of the seam cell network was not as successful in revealing reproducible differences in SCN between isolates. For example, putative null mutations in *bro-1*, the CBFβ homologue and binding partner of RNT-1/Runx, or in the fusogen *eff-1*, led to comparable changes in SCN between pairs of wild isolates (Supplementary Fig. 1c, d). Based on these results,

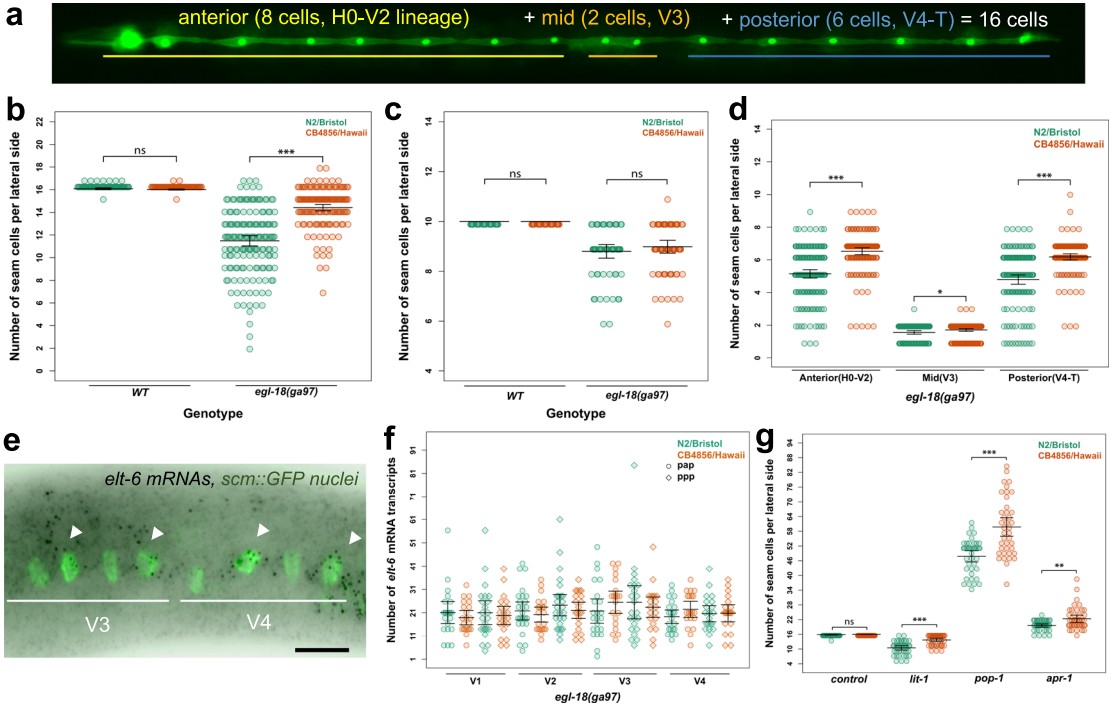

**Fig. 1 The egl-18(ga97) mutation results in lower seam cell number in the N2 background compared with CB4856. a** Image depicting the stereotypic seam cell distribution in wild-type animals at the end of larval development, where eight cells are positioned anterior to the vulva (H0–V2), two cells are near the vulva (V3 lineage) and six cells are found posterior to it (V4–T). **b** Average terminal seam cell number (SCN) is significantly lower in N2 (green) than CB4856 (orange) carrying the egl-18(ga97) mutation (one-way ANOVA $F$ (1, 310) = 115.02, $p < 2.2 \times 10^{-16}$, $n \geq 150$), but is not different in wild-type isolates (one-way ANOVA $F$ (1, 142) = 0.13, $p = 0.13$, $n = 72$ independent animals for wild-type and $n = 150$ or 162 independent egl-18 mutants in N2 and CB4856 respectively). **c** No statistically significant difference in SCN is found between N2 and CB4856 wild-type isolates or egl-18(ga97) mutants at the end of L1 stage with one-way ANOVA ($p > 0.32$, $n = 60$ and 64 independent egl-18 mutant animals in N2 and CB4856 and $n = 25$ independent animals for wild-type). **d** Seam cell number comparison between egl-18(ga97) mutants in N2 and CB4856 when seam cells are classified based on their position (anterior, mid, posterior). Differences in SCN are significant between the two isolates for all classes based on one-way ANOVA ($F$ (1, 310) = 66.41, 4.87 and 61.78, *$p < 0.03$ and ***$p < 0.0001$, $n \geq 150$ independent animals). **e** Representative smFISH image showing elt-6 expression in posterior seam cells in an egl-18 mutant at the late L2 stage. Seam cell nuclei are labelled in green due to scm::GFP expression and black spots correspond to elt-6 mRNAs. A similar pattern of expression was observed in three independent experiments. Scale bar is 20 μm. **f** Quantification of elt-6 expression by smFISH in N2 ($n = 186$ cells) and CB4856 ($n = 164$ cells) carrying the egl-18(ga97) mutation. No statistically significant difference in the mRNA was found with one-way ANOVA ($F$ (1, 348) = 0.13, $p = 0.72$). **g** SCN quantification in N2 and CB4856 strains upon knockdown of lit-1, pop-1, and apr-1. There is a significant effect upon knockdown of lit-1, pop-1 and apr-1 RNAi using one-way ANOVA ($F$ (1, 78) = 32.71, 30.27, 7.3 and 1.37, ***$p < 0.0001$ or **$p < 0.001$, $n = 40$ independent animals). Error bars in **b–d**, **f**, **g** indicate 95 % confidence intervals around the mean. Source data are provided as a Source Data file.

we decided to focus on understanding the difference in phenotype expressivity driven by mutations in egl-18(ga97) between N2 and CB4856, two isolates for which genomic differences are well-characterised[21,22].

As EGL-18 acts to maintain seam cell fate throughout development[18,19], the difference in phenotype expressivity between N2 and CB4856 egl-18(ga97) mutants could arise from changes during embryonic or post-embryonic development. To distinguish between these two possibilities, we compared SCN between egl-18(ga97) mutants in the two isolates at the end of the L1 stage. We found that both isolates showed a comparable SCN at this early stage (Fig. 1c). We then compared the spatial distribution of seam cells at the L4 stage between the two isolates carrying the egl-18(ga97) mutation. Here, we found significantly higher seam cell counts in the CB4856 background throughout the body axis (Fig. 1d). There are multiple ways to explain the higher terminal SCN in the egl-18 mutant in CB4856 based on our knowledge of the seam cell lineages. The first possibility involves the occurrence of additional symmetric divisions, either owing to symmetrisation of normally asymmetric cell divisions, where anterior cell daughters maintain the seam cell fate instead of differentiating into hypodermis, or because of ectopic symmetric cell divisions that expand the seam cell pool.

Alternatively, an increase in seam cell fate maintenance could reduce the tendency for precocious cell differentiation in the egl-18(ga97) mutants, and thus increase terminal SCN independently of cell division. Despite phenotyping large numbers of egl-18 (ga97) animals, we never observed tightly clustered seam cell nuclei, which are indicative of seam cell symmetrisation events[12]. Taken together, these results suggest that egl-18(ga97) mutants may show higher terminal SCN in the CB4856 background compared to N2 due to higher seam cell fate maintenance across all lineages.

One explanation behind mutation buffering is compensation within a gene network, for example, if loss-of-function in one gene can be compensated by an increase in the expression of a closely related paralogue[23–25]. In this case, elt-6 could compensate for the loss of its paralogue egl-18 to a different level between CB4856 and N2. It is of note that such genetic compensation could only occur in this case in the form of gene expression regulation in *trans* to the elt-6 locus, as opposed to regulation in *cis*. This is because egl-18 and elt-6 are tightly linked so the full elt-6 locus was transferred from N2 to CB4856 together with the egl-18(ga97) mutation during the genetic introgression. We used single-molecule fluorescent in situ hybridisation (smFISH) to compare gene expression between egl-18(ga97) mutants in N2

and CB4856 and found that although the expression of *elt-6* was higher in the presence of the *egl-18* mutation, there was no significant difference in *elt-6* or *egl-18* expression between the two isolates in a wild-type or *egl-18(ga97)* mutant background (Fig. 1e, f and Supplementary Fig. 2a, b). This finding suggests that the difference in phenotype expressivity between N2 and CB4856 *egl-18(ga97)* mutants is unlikely to be dependent on changes in *elt-6* expression.

Wnt signalling has a role in seam cell development and natural variation in Wnt requirement for intestinal development has recently been reported[26]. Therefore, we hypothesised that the difference in phenotype expressivity between the two isolates might reflect changes in the Wnt pathway. To test this hypothesis, we targeted genes involved in this signalling pathway by RNAi and compared seam cell counts between N2 and CB4856. We found that RNAi against *lit-1/NLK* decreased SCN in both isolates, but N2 was more sensitive to lose seam cells than CB4856 (Fig. 1g). CB4856 is known to harbour variation that makes this strain insensitive to germline RNAi[27–29], and seam cell RNAi was found to be less effective in CB4856 than N2 (Supplementary Fig. 2c). Therefore, we cannot rule out that the observed difference upon *lit-1* RNAi is due to changes in RNAi sensitivity between the two isolates. RNAi knockdown of *apr-1/APC* and *pop-1/TCF* increased SCN in both isolates compared to control treatment as expected. However, CB4856 was found to be more sensitive to gain seam cells than N2 (Fig. 1g), which suggested an inherent difference between the two isolates in the outcome of Wnt-related perturbation. We compared Wnt pathway activity between wild-type N2 and CB4856 animals using an established marker reflecting POP-1 binding in cells where Wnt signalling is activated[30], but found no reproducible difference (Supplementary Fig. 2d). Finally, the difference in phenotype expressivity between the two isolates was masked upon RNAi of *elt-6* or *pop-1* (Supplementary Fig. 2e), highlighting a requirement for Wnt signalling for the difference to manifest. Taken together, we conclude that the difference in phenotype expressivity between N2 and CB4856 *egl-18(ga97)* mutants is likely to reflect changes that potentiate seam cell fate maintenance by Wnt-signalled cells.

Animals carrying the *egl-18(ga97)* mutation in both isolates display aberrant vulval phenotypes, such as a protruding vulva and egg-laying defects. This is because EGL-18 is known to be required for maintenance of vulval precursor cell fate by inhibiting cell fusion to hypodermis[31,32]. We tested whether vulval cell fate induction is different between N2 and CB4856 animals carrying the *egl-18(ga97)* mutation. We found no statistically significant difference between the two isolates (Supplementary Fig. 2f). We finally tested whether a previously reported polymorphism in the N-acetyltransferase *nath-10* between N2 and other isolates including CB4856, which has been shown to influence the outcome of EGF/Ras mutations in vulva development[6], was able to modify SCN. We found that the CB4856 *nath-10* polymorphism is not sufficient to modify the phenotype expressivity of the *egl-18(ga97)* mutants in the N2 background (Supplementary Fig. 2g). Taken together, these results suggested that the difference in phenotype expressivity between N2 and CB4856 *egl-18(ga97)* mutants is specific to the seam and does not rely on previously identified genetic variation that is already known to affect epidermal development.

**Multiple QTLs influence the phenotype expressivity between the two isolates**. To discover the genetic basis underlying the difference in SCN between N2 and CB4856 carrying the *egl-18 (ga97)* mutation, we used a quantitative genetic approach. We produced 116 homozygous recombinant inbred lines (RILs), which consisted of shuffled parental genomes, whereas carrying the *egl-18(ga97)* mutation and the *scm::GFP* marker to be able to visualise the seam cells (Fig. 2a). Upon phenotypic characterisation of the resulting RILs, we found that the SCN distribution was continuous, with most RILs displaying an average SCN that was intermediate between the two parents (Fig. 2b). Independent phenotyping of the generated RILs showed good reproducibility, pointing to a genetic basis underling the difference (Supplementary Fig. 3a). We also observed transgressive segregation, with some RILs showing a SCN that is higher than CB4856 or lower than N2, indicating that multiple genetic loci are likely to affect this phenotype.

We used bulked segregant analysis combined with whole-genome sequencing to map the genetic loci involved[33,34]. In this approach, the extremes on each side of the phenotypic distribution that resemble the parental phenotype (i.e., they show either low average SCN like N2 or high average SCN like CB4856) are pooled and sequenced together as a bulk (Fig. 2b). According to the null hypothesis, there should be no statistically significant differences in the SNP frequencies between the low and high-SCN bulk samples for genomic regions that do not influence the phenotype. However, deviations are expected for genomic regions that influence positively or negatively the phenotype. Interestingly, we observed significant deviation in SNP frequencies on multiple chromosomes, suggesting the presence of multiple QTLs (Fig. 3). In particular, we identified at least four significant QTLs on chromosomes II, III, V, and X that are likely to modulate the phenotypic outcome of the *egl-18(ga97)* mutation (Fig. 3a–f). In most cases, the high bulk contained CB4856 alleles in these QTL regions, except for the right arm of chromosome V where the high bulk contained N2 alleles. Because of the complexity of the experimental design, impaired resolution was anticipated in certain chromosomal areas that were exclusively derived from N2 in all RILs. For example, RILs in both groups harboured a short region on the left arm on chromosome IV from N2, which corresponds to the region where the introgressed *egl-18(ga97)* mutation resides (Fig. 3d). Also RILs in both groups had a large portion of chromosome V from N2, which is where the *scm::GFP* transgene is inserted (Fig. 3e). Finally, a region on the left arm of chromosome I was mostly derived from N2 (Fig. 3a), because this region harbours a previously known genetic incompatibility between the two parental isolates[35,36].

**The identified QTLs harbour cryptic genetic variation affecting seam cell development through the Wnt pathway**. To investigate the putative effect of these QTLs on seam cell development, we created near isogenic lines (NILs) in the wild-type and *egl-18 (ga97)* mutant N2 background. To break-down the genetic composition of individual RILs in the two bulks and study their association with the SCN phenotype, we took advantage of known indels between the two parental strains to design markers for genotyping around the QTL regions[22]. Based on initial genotyping results, we used RIL28 to create NILs as this was one of the lines in the high bulk showing the highest average SCN while it contained shorter fragments from CB4856 on both chromosomes II and III (Supplementary Fig. 3b). By isolating in N2 the independent fragments derived from CB4856 chromosomes II, III and X, we found that fragments on chromosomes II and III individually increased SCN compared to the *egl-18(ga97)* mutant phenotype in N2 (Fig. 4a). The fragment on chromosome X was not sufficient on its own to increase the average SCN of the *egl-18 (ga97)* mutant in N2, but it was able to act together with the QTL on chromosome III to increase the average SCN further (Fig. 4a). Interestingly, lines carrying a combination of two or three QTLs together showed a SCN phenotype very comparable to *egl-18 (ga97)* mutants in CB4856, suggesting that the identified QTLs

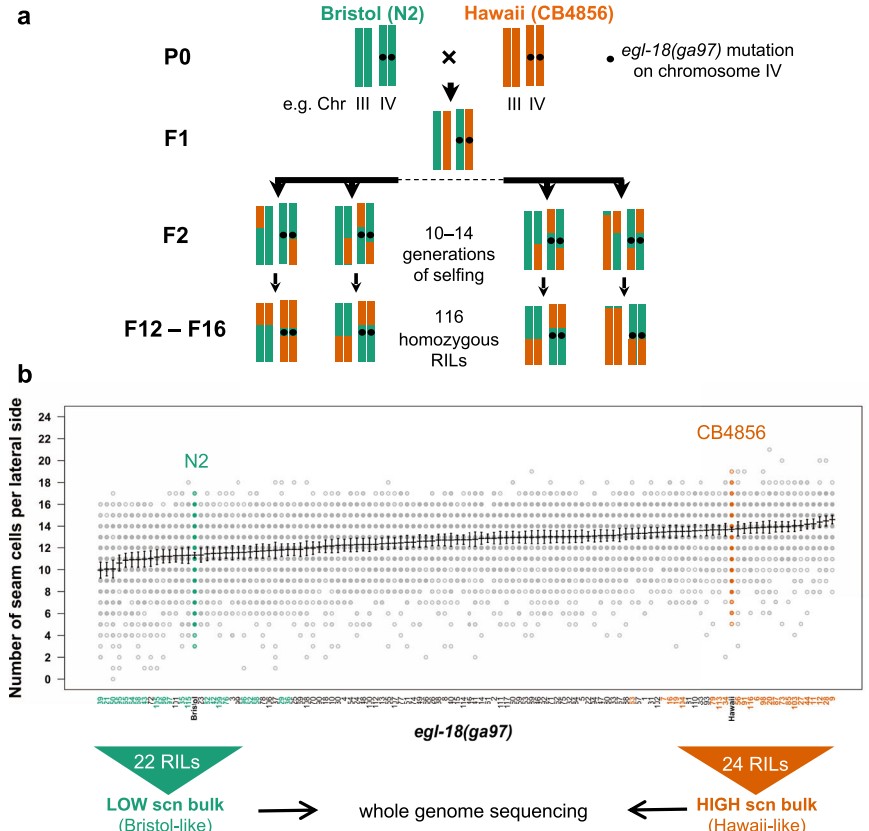

**Fig. 2 Generation and phenotypic analysis of recombinant inbred lines. a** Generation of RILs between CB4856 and N2 carrying the *egl-18(ga97)* mutation. Two chromosomes (III and IV) are shown for simplification. **b** Seam cell number quantification in 116 recombinant inbred lines, ranked from lowest to highest average seam cell number. Parental strains are shown in green (N2) and orange (CB4856). The two extremes (low-bulk and high bulk) of the phenotypic distribution were pooled for whole-genome sequencing. Error bars indicate average SCN ± 95% confidence intervals, $n = 77$ independent animals for RIL95, $n = 80$ for all other RILs and $n = 520$ independent animals for the N2 and CB4856 controls.

capture a large proportion of the genetic variation influencing this phenotypic trait (Fig. 4a). Furthermore, the same QTLs did not alter wild-type seam cell development alone or in combination (Fig. 4b, c), so their influence was only manifested by modifying the phenotypic outcome of the *egl-18(ga97)* mutation. These results thus highlight the presence of complex genetic variation in *C. elegans*, which remains cryptic in wild-type condition, but is able to influence seam cell development upon genetic perturbation.

We then tested whether the identified QTLs could also influence the phenotypic outcome in other sensitised conditions in the absence of mutations in *egl-18*. Given that the wild-type parental strains responded differently to knockdown of Wnt signalling components, we used RNAi to knockdown the expression of *pop-1* in strains containing combinations of the QTL fragments (on chromosomes II, III and X) from CB4856 in a wild-type N2 background. We found that *pop-1* RNAi led to significantly higher SCN in animals carrying CB4856 QTL regions compared to N2 (Fig. 4c). This finding indicates that the identified QTLs may promote seam cell fate by potentiating the Wnt signalling pathway.

**Natural variation in *hsp-110* contributes to the difference in phenotype expressivity between the two *C. elegans* isolates**. To narrow down the genomic interval of the two major QTLs on chromosomes II and III, we screened for further recombinants after crossing our NILs to N2 and screening for lines in which the presence of a defined CB4856 fragment was sufficient to increase SCN of *egl-18(ga97)* mutants in the N2 background to near

CB4856 levels. With regard to chromosome II, we derived two NILs (strain MBA944 and MBA846), which largely rescued the expressivity of the *egl-18(ga97)* mutant in N2, whereas they contained mostly distinct genomic regions with a small over-lapping region in the middle of ~0.14 Mb (Supplementary Fig. 4). Non-overlapping regions in MBA944 and MBA846 contained 106 and 116 genes harbouring natural variation, with seven genes in the overlap (*dgk-5, del-10, T28D9.1, abch-1, utp-20, wrn-1, C56C10.9*). To narrow down these candidates further, we reasoned that genes within the QTLs responsible for the difference in phenotype expressivity between N2 and CB4856 *egl-18(ga97)* mutants would modify SCN when targeted by RNAi in an *egl-18 (ga97)* mutant background. However, we found no difference in SCN upon downregulating the 7 genes in the overlap in the *egl-18 (ga97)* background in N2 or N2 carrying the QTLs on chromosome II, III and X from CB4856 (Supplementary Fig. 5a). These results are therefore more compatible with the possibility of two independent QTLs on chromosome II. Consistent with this hypothesis, we focused on *egl-27* and *dsh-2*, which are located at the two opposite sides of the overlap and harbour natural genetic variation (Supplementary Fig. 4c). These genes are also known to play a role in seam cell patterning[37–39]. We found that RNAi knockdown of *egl-27* and *dsh-2* decreased SCN in the *egl-18(ga97)* mutant background (Supplementary Fig. 5a). Therefore, natural variation in these candidates may contribute to the difference in the phenotypic outcome of the *egl-18(ga97)* mutation between the two isolates.

Concerning chromosome III, we found several candidates located within the broad QTL region that led to SCN decrease in

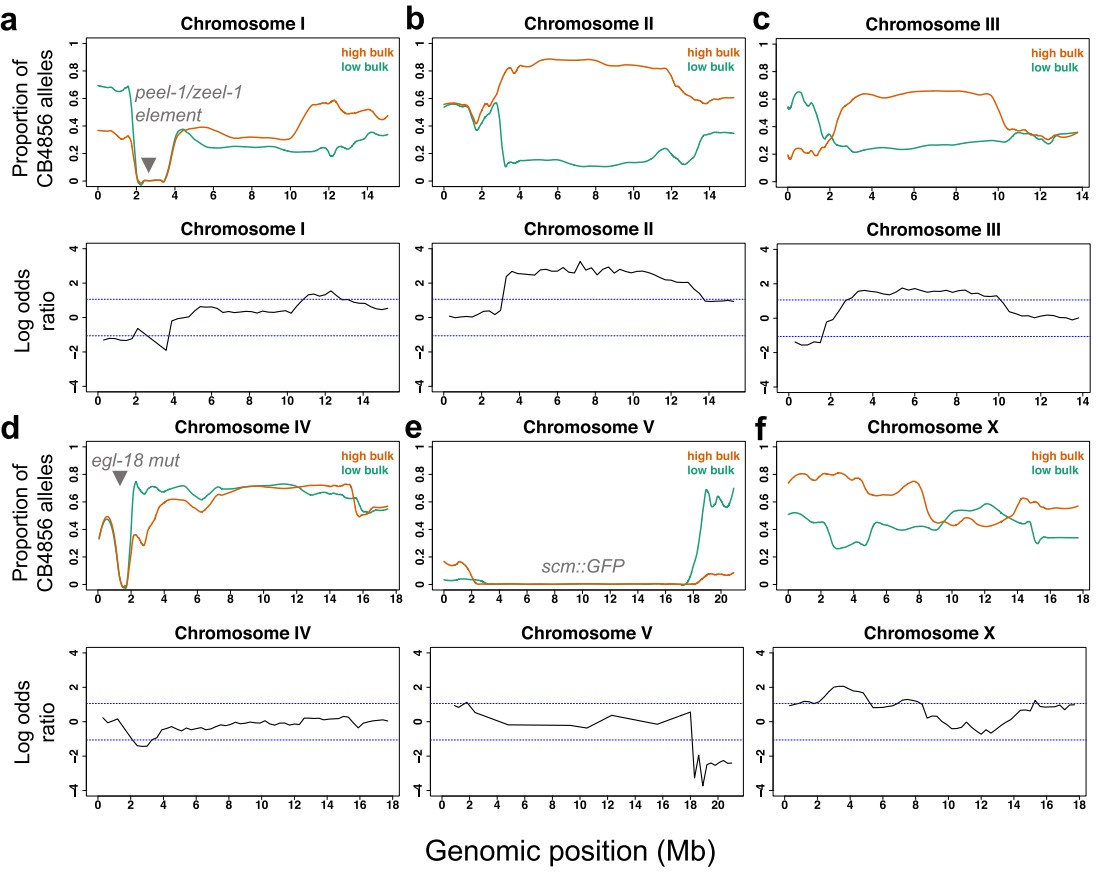

**Fig. 3 Bulk segregant analysis of recombinant inbred lines. a–f** Proportion of CB4856 SNPs in the sequencing reads of the low-seam cell number (SCN) bulk and high-SCN bulk groups is shown along the six chromosomes, from I **a** to X **f**. SNP frequencies in low-bulk and high bulk are shown in green and orange fitted curves, respectively. The curves represent locally weighted scatterplot smoothing (LOESS) regression lines from the allele frequencies at known SNP positions along the chromosomes with a span parameter of 0.1. Log-odds ratio of average SNP frequencies in non-overlapping 300 kb windows along the six chromosomes is also shown. The blue dashed lines indicate the thresholds for statistical significance for log-odd ratios at $\alpha = 0.05$. The positions of the *peel-1/zeel-1* element, *egl-18(ga97)* mutation and *scm::GFP* transgene integrations are shown in grey. Negative LOD scores are obtained when the low-SCN bulk has a higher proportion of CB4856 SNPs compared to the high-SCN bulk.

the *egl-18(ga97)* mutant when these were targeted by RNAi (Supplementary Fig. 5b). We were then able to narrow down the QTL to a smaller genomic interval of ~1.14 Mb, which contained 59 genes in total with known genetic variation (Fig. 5a, b, Supplementary Figure 4d and Supplementary Data 1), with two strong candidates based on our RNAi experiments. First, *sor-1*, a species-specific polycomb group protein that is involved in epigenetic silencing of Hox genes[40]. Genetic variation in this gene involves one non-synonymous substitution and one adjacent synonymous mutation, both in exon 7. The second candidate was *hsp-110*, a homologue of HSPA4 and member of the HSP70 gene superfamily[41,42]. Genetic variation in *hsp-110* involves a 3 bp in-frame deletion in exon 5 in CB4856, which deletes a conserved amino acid (D474) (Fig. 6a).

To validate these variants, we used CRISPR-mediated genome editing to engineer them into N2 and assess their impact on seam cell development both in a wild-type and in an *egl-18* mutant background. Interestingly, we found that the CB4856 *hsp-110* allele significantly increased SCN when introduced into an *egl-18 (ga97)* mutant background in N2 (Fig. 6b), whereas no effect was observed in a wild-type background (Fig. 6c). Instead, we observed no effect on SCN when we introduced the CB4856 *sor-1* polymorphisms in a wild-type or *egl-18* mutant in the N2 background (Fig. 6b–c).

We used smFISH to study the expression of *hsp-110* and found evidence for expression around seam cell nuclei, with no apparent

difference between N2 and CB4856 (Fig. 6d, e), suggesting that HSP-110 is likely to act cell-autonomously to modify seam cell behaviour. Knockdown of *hsp-110* had no effect in wild-type, but decreased seam cell counts further in strains carrying strong loss-of-function alleles of *egl-18* (Supplementary Fig. 6), which led us to hypothesise that HSP-110 is likely to promote seam cell fate and this effect may be enhanced in a CB4856 background carrying the polymorphic *hsp-110* allele. To test this hypothesis, we performed *hsp-110* RNAi in sensitised N2 backgrounds carrying mutations in Wnt components, such as the Wnt ligand *egl-20*, the Frizzled receptor *lin-17* and the canonical β-catenin *bar-1*. Interestingly, we found that while *hsp-110* knockdown did not modify the *egl-20* or *lin-17* mutant phenotype, it significantly decreased SCN in the *bar-1* mutant (Fig. 6f). Conversely, we found that the CB4856 allele of *hsp-110* amplified the increase in SCN observed upon *pop-1* knockdown in N2 (Fig. 6g). Taken together, these results suggest that natural genetic variation in a conserved heat shock protein contributes through the Wnt pathway to the difference in phenotype expressivity between CB4856 and N2 *egl-18(ga97)* mutants.

## Discussion

Previous work in various models has shown that the phenotypic effect of mutations can vary depending on the genetic background[4,43,44]. For example, the severity of null mutations in mice can show opposing effects depending on the strain[4], indicating

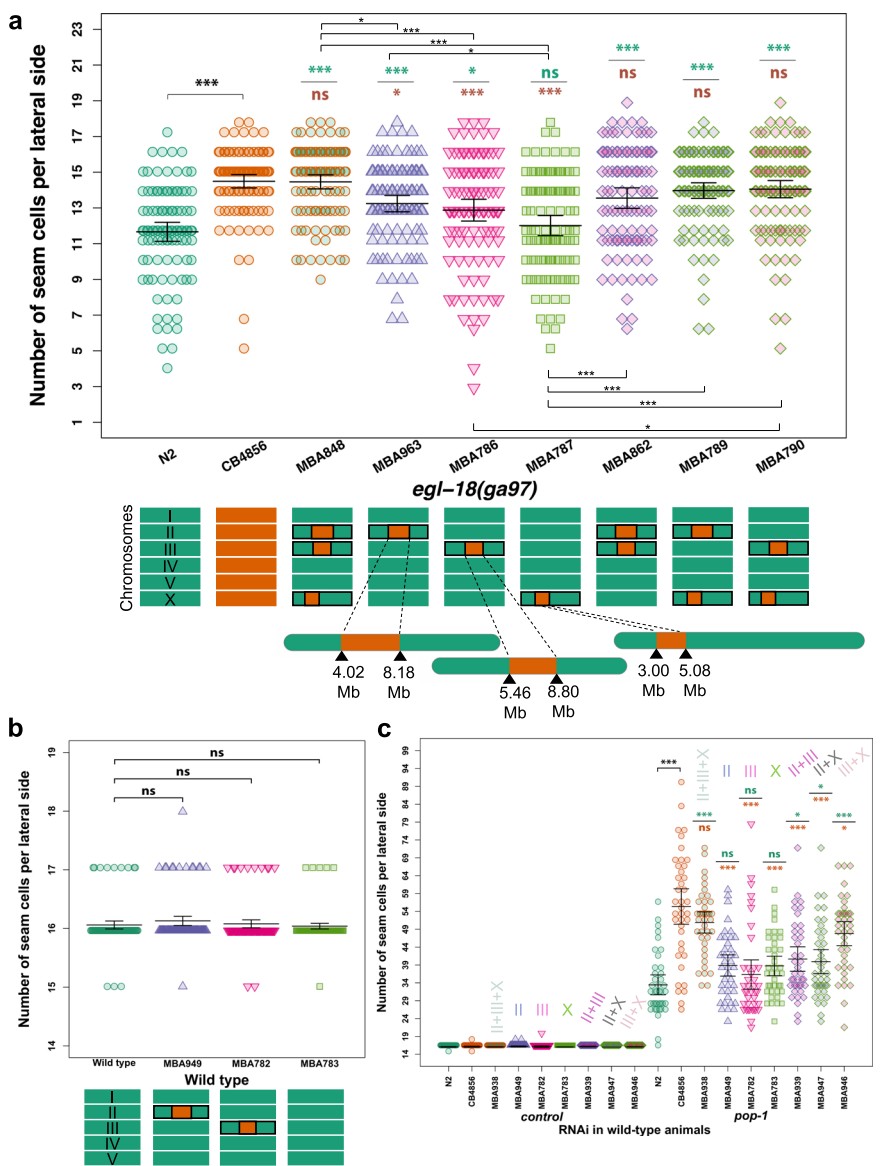

**Fig. 4 Phenotypic analysis of near isogenic lines identifies major QTLs. a** Seam cell number in near isogenic lines containing individual and combinations of the identified QTLs from CB4856 into the *egl-18(ga97)* N2 background. One-way ANOVA showed that SCN was significantly affected by the strain ($F_{(8, 905)} = 16.55$, $p < 2.2 \times 10^{-16}$, $n \geq 100$ independent animals). Stars show significant differences by post hoc Tukey HSD (***$p < 0.001$ or *$p < 0.05$). Note that QTLs on chromosomes II and III, but not on the X, were sufficient to increase seam cell number in the *egl-18(ga97)* mutant in the N2 background. Cartoon of chromosomes below the graph depict the genotype of the strain. **b** SCN counts in near isogenic lines containing individual QTLs in a wild-type background. One-way ANOVA shows no statistically significant differences in SCN of NILs compared with wild-type with no QTLs ($F_{(3, 396)} = 1.34$, $p = 0.26$, $n = 100$ independent animals). **c** *pop-1* RNAi in near isogenic lines in a wild-type background containing individual and combinations of the identified QTLs. The same colour code is used as in **a** to define different combinations of QTLs. A significant effect of strain was found on SCN upon *pop-1* RNAi using one-way ANOVA ($F_{(8, 351)} = 17.85$, $p < 2.2 \times 10^{-16}$, $n = 40$ independent animals). Error bars in **a**–**c** indicate 95% confidence intervals around the mean. Comparisons against N2 and CB4856 are shown with green and orange stars, respectively, in **a** and **c** and represent $p$ values by one-way ANOVA followed by post hoc Dunnett's multiple comparison test (*** $p < 0.001$ or * $p < 0.05$). Source data are provided as a Source Data file.

that the genetic background can influence the genotype-to-phenotype relationship in a complex manner[45]. In *C. elegans*, genetic variation can be cryptic in wild-type conditions and influence phenotypes ranging from gene expression to development of physiology in complex ways upon perturbation[29,46,47]. Cryptic genetic variation can also modify phenotype expressivity as shown for a lab evolved non-synonymous polymorphism in *nath-10* in N2, which alters vulval cell fate induction in the presence of a mutation in the epidermal growth factor receptor[6].

We report that the phenotypic outcome of the *egl-18(ga97)* mutation in seam cells is milder in CB4856 compared with the

laboratory reference strain N2. GATA factors, similar to EGL-18, are implicated in disease such as blood and heart disorders, with often variable phenotypic outcomes in human patients[17,48]. Using a quantitative genetic approach, we discover multiple QTLs that modify SCN in an *egl-18(ga97)* mutant background, acting independently, as well as in combination. We pin down the molecular basis of one of these QTLs and demonstrate that a deletion of a single amino acid (D474) in the heat shock protein HSP-110, which has occurred during the evolution of the CB4856 strain, is sufficient to increase the average SCN when introduced into N2. HSP-110 proteins form a distinct branch of

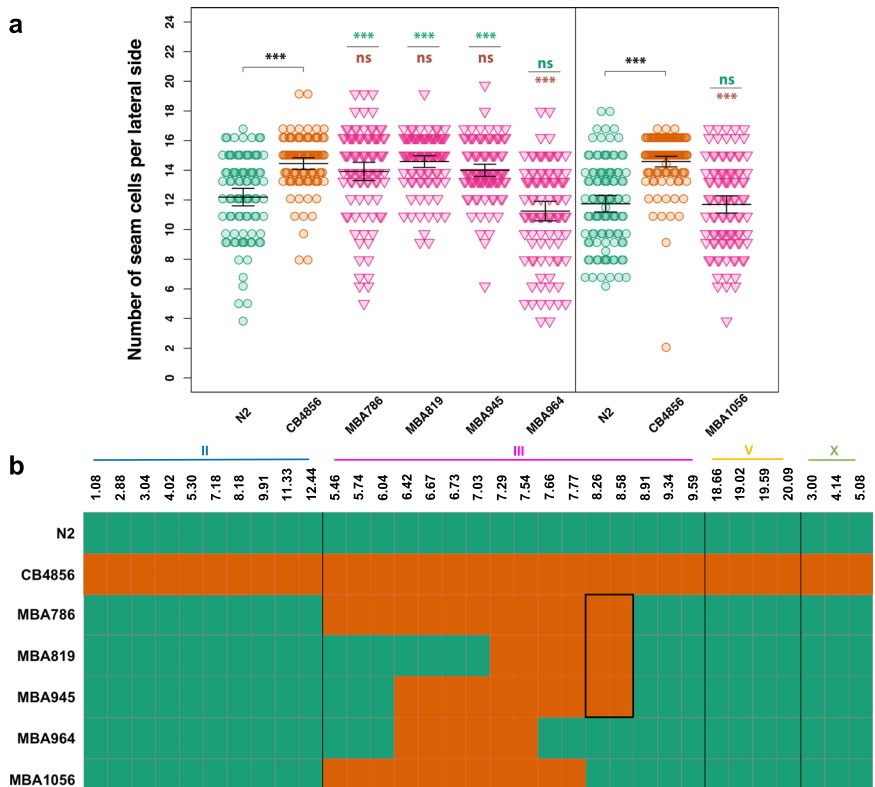

**Fig. 5 Genotype-to-phenotype analysis of NILs carrying genomic fragments of chromosome III from CB4856. a** Seam cell number in near isogenic lines containing various fragments of chromosome III from CB4856 in the *egl-18(ga97)* N2 background. Vertical black line in the graph distinguishes between two independent experiments. In both experiments, one-way ANOVA showed that SCN was significantly affected by the strain ($F$ (5, 556) = 26.58, $p <$ $2.2 \times 10^{-16}$; $F$ (2, 313) = 43.31, $p < 2.2 \times 10^{-16}$ respectively, $n \geq 90$ independent animals). Error bars indicate 95% confidence intervals around the mean and \*\*\*$p < 0.0001$ corresponds to significant differences by post hoc Tukey HSD compared with N2 (green) and CB4856 (orange) respectively. **b** Genotyping of NILs carrying genomic fragments of chromosome III from CB4856 using genetic markers. Green and orange tiles represent N2 and CB4856 genomes, respectively. Highlighted box in the middle indicates the genomic region that is shared in the NILs in which phenotype expressivity is converted from N2 to CB4856 levels. Source data are provided as a Source Data file.

the Hsp70 gene superfamily[41,49,50], which includes conserved proteins that assist in protein folding and protect cells from stress[51]. HSP-110 members are thought to assist in solubilisation of protein aggregates through nucleotide exchange activity rather than acting as canonical ATPase chaperones[52,53]. HSP-110 has not been studied before in the context of *C. elegans* development. Previous reports have demonstrated a role for HSP-110 in survival upon heat shock[52], as well as recovery from thermal stress in plant pathogenic nematodes, where evolution in *hsp-110* has occurred in *Globodera* species through gene duplication[54].

Molecular chaperones, like Hsp-90, can modulate the genotype-to-phenotype relationship[14,55]. Hsp-90-mediated buffering of protein folding has been thought to influence the variable expressivity of human disease[56]. In the seam, RNAi knockdown of *hsp-90* or DnaJ chaperones leads to increased variability in SCN[10,57]. Differences in heat shock protein expression have also been shown to affect the incomplete penetrance of mutations[23,58]. Our work highlights that natural variation in conserved heat shock proteins can shape phenotype expressivity. It is of note that mutations in HSP-110 in human patients correlate with variable responses to drug chemotherapy[59], so it is conceivable that mutations in conserved HSP proteins may influence the phenotypic outcome of disease-causing loci.

EGL-18 and ELT-6 transcription factors act redundantly to promote seam cell fate[18], with EGL-18 playing the major role in both isolates. Genetic compensation has been shown to occur in various organisms including *C. elegans* as a mechanism to buffer against null mutations to protect from developmental failure[23–25].

Although we did find an increase in *elt-6* expression in the *egl-18* mutant background compared with wild-type suggestive of compensation, there was no significant difference in *elt-6* expression between the two isolates with or without the *egl-18* mutation. Therefore, it is unlikely that differential regulation of *elt-6* at the mRNA level contributes to the difference in phenotype expressivity. Given the tight linkage between *egl-18* and *elt-6*, we cannot formally rule out that the less-severe outcome of the *egl-18* (*ga97*) mutation in CB4856 can only be seen in the presence of the *elt-6* locus from N2. However, this is unlikely because differences in seam cell counts were also observed independently of *egl-18/elt-6* introgression, for example, upon *pop-1* RNAi in the CB4856 background compared to N2. We therefore propose a model wherein changes in the seam cell regulatory network through accumulation of natural variation, including the amino-acid deletion found within HSP-110 in CB4856, potentiate the Wnt signalling pathway to promote seam cell fate (Fig. 7). The exact mechanistic relationship between HSP-110 and the Wnt pathway remains to be identified. Based on the position of the deleted amino-acid within the substrate binding domain of HSP-110, we speculate that some altered interaction with a protein partner may stabilise HSP-110 in CB4856 and thus potentiate Wnt signalling. It is of note that a crosstalk between related Hsp proteins and Wnt signalling has been proposed in other contexts to act at the level of β-catenin activation[60], which is in keeping with the synthetic interaction we observed upon knockdown of *hsp-110* in the presence of a *bar-1* null allele. This interaction also reveals a new role for *bar-1* in the seam cells, which is presumably

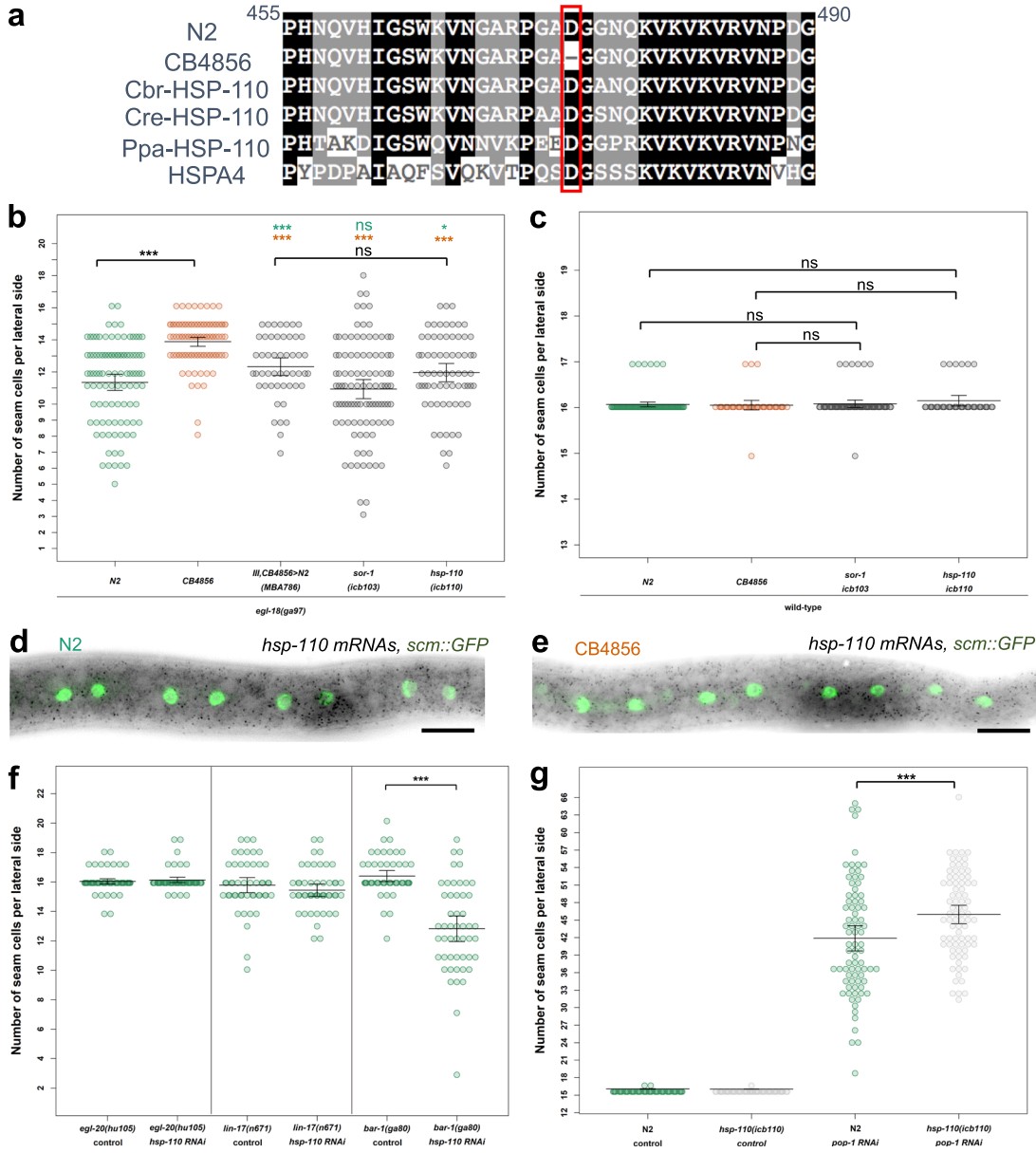

**Fig. 6 Natural variation in *hsp-110* contributes to the difference in phenotype expressivity between N2 and CB4856. a** Alignment of the HSP-110 amino-acid sequence from *C. elegans* (N2 and CB4856), *C. remanei (Cre)*, *C. brenneri (Cbr)*, *P. pacificus (Ppa)*, and *H. sapiens* (HSPA4) around the amino-acid deletion in CB4856 (highlighted in red). **b–c** Comparison of seam cell counts between animals carrying the CB4856 variants of *hsp-110* and *sor-1* in N2 in the *egl-18(ga97)* (**b**) or wild-type background (**c**); *n* > 68 independent animals in (**b**) and *n* > 40 independent animals in **c**. Comparison is shown to parental isolates or the introgression of the QTL on chromosome III as a reference. Significance is tested by one-way ANOVA followed by post hoc Tukey HSD and differences are shown compared with N2 (green stars) and CB4856 (orange stars) respectively, *** corresponds to *p* < 0.001. **d–e** Representative smFISH images showing *hsp-110* expression in divided seam cells in N2 or CB4856 at the late L2 stage. Seam cell nuclei are labelled in green due to *scm::GFP* expression and black spots correspond to *hsp-110* mRNAs. A similar pattern of expression was observed in two independent experiments. Scale bar is 20 μm. **f** SCN quantification in N2 carrying mutations in Wnt components upon knockdown of *hsp-110*, (*n* > 49 independent animals for all strains, significance is reported with a two-tailed Welch *t* test; *p* = 0.31 for *lin-17(-)*, *p* = 0.43 for *egl-20(-)* and *p* < 0.0001 for *bar-1(-)* mutants). Vertical lines mark independent experiments. **g** SCN quantification in N2 or N2 carrying the CB4856 *hsp-110* allele upon knockdown of *pop-1* (*n* = 80 independent animals). Significant change is found upon *pop-1* RNAi treatment (*p* = 0.0008 by one-way ANOVA followed by post hoc Tukey HSD). Error bars in **b–c**, **f–g** indicate 95 % confidence intervals around the mean. Source data are provided as a Source Data file.

masked by other b-catenins acting redundantly (Gleason and Eisenmann, 2010). The tractability of the *C. elegans* seam cell model allows therefore to identify and study genetic modifiers of the Wnt signalling pathway. Cryptic genetic variation in the requirement for Wnt signalling has also been revealed in the context of gut differentiation[26], suggesting that natural variation may broadly influence Wnt-dependent gene regulatory networks in *C. elegans*.

The amino-acid deletion in *hsp-110* is specific to the CB4856 isolate of *C. elegans* (CeNDR release 20180527). CB4856 belongs to a group of highly polymorphic strains found around geographically isolated Hawaiian islands. These isolates are thought to represent ancestral genetic diversity because they contain approximately three times more diversity than the non-Hawaiian populations[11,61,62]. It is of note that we recently reported that a rise in culture temperature, from 20 to 25 °C, increases SCN

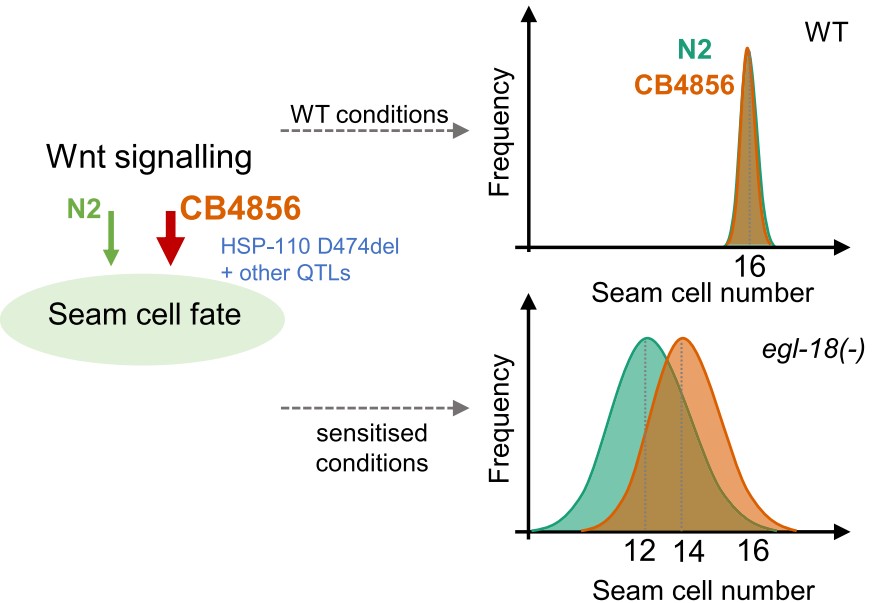

**Fig. 7 Model explaining the difference in the outcome of the *egl-18(ga97)* mutation between N2 and CB4856.** Seam cell number distribution is shifted in *egl-18* mutants in CB4856 (orange) compared with N2 (green) towards a higher average. Genetic variation in *hsp-110* and other loci within the identified QTL regions in CB4856 may potentiate Wnt signalling and thus reinforce seam cell fate acquisition. This effect is only revealed in sensitised conditions, such as upon *egl-18* loss-of-function or *pop-1* RNAi.

through symmetrisation of normally asymmetric cell divisions in many *C. elegans* isolates, except for CB4856[12]. It is therefore intriguing that CB4856 appears to harbour loci that both reinforce seam cell fate maintenance and suppress ectopic seam cell fate acquisition, which again highlights that genetic variation can influence developmental phenotypes in a complex manner.

Although natural variation in *hsp-110* contributes to the difference in phenotype expressivity, we also revealed a complex genetic architecture. When it comes to other loci involved, our results suggested two strong candidates for the QTL on chromosome II, that is the Dishevelled protein *dsh-2* and *egl-27*. There are three Dishevelled proteins (MIG-5, DSH-1, and DSH-2) in *C. elegans*. DSH-2 and MIG-5 have been shown to act redundantly to regulate seam cell fate by positive regulation of nuclear SYS-1 β-catenin levels[37]. EGL-27 is an ortholog of human RERE (arginine-glutamic acid dipeptide repeats protein) and mutations in *egl-27* have been associated with seam cell defects leading to an increase in SCN[38,39]. Chromatin-modifying regulators, such as EGL-27, have been proposed to represent highly connected hubs in *C. elegans* genetic interaction networks[63], so they may influence the phenotypic outcomes of multiple mutations in factors belonging to several unrelated pathways. Future work is required to understand the significance of these natural variants and their possible connection to the Wnt signalling pathway.

## Methods

**Strains and genetics.** *C. elegans* was maintained on a lawn of *Escherichia coli* strain OP50 seeded on Nematode Growth Medium (NGM) according to standard procedures[64]. *C. elegans* larvae were synchronised by bleaching gravid hermaphrodites and washing the eggs twice with M9 buffer. All strains used in this study are listed as Supplementary Data 2. Mutations in *egl-18* were introgressed into wild-type strains MBA256 and MBA19 (CB4856 and JU2007, respectively, carrying the *wIs51* transgene)[12] in a two-step cross. In the first step, MBA256 and MBA19 males were crossed to *egl-18(-)* hermaphrodites carrying the *wIs51* transgene. In the second step, F1 males from the previous cross were crossed to wild isolate hermaphrodites. F2 animals from the second cross that were egg-laying defective were isolated and the two-step cross was repeated five times to produce 10× backcrossed strains. The process was repeated for the introgression of *eff-1(icb4)* into the MBA19 background. The *eff-1(icb4)* mutation leads to a premature stop codon (Q148STOP).

**RNA interference.** Adult gravid animals (wild-type or *egl-18* mutants) were bleached and eggs were added onto RNAi plates to hatch and feed on bacteria expressing double-stranded RNA. For small scale experiments, *egl-18* mutant adults were also punctured with an injection needle to release eggs. Bacterial clones originate from the Ahringer and Vidal libraries, or were custom made (*wrn-1*, *utp-20*). Bacterial clones were grown overnight in liquid LB medium with 50 μg/ml ampicillin and 12.5 μg/ml tetracycline. The bacterial cultures were seeded onto filter-sterilised isopropyl β-D-1-thiogalactopyranoside-containing RNAi plates and allowed to dry for 2–4 days at room temperature before use. Custom RNAi clones were made by amplifying gene fragments for a gene of interest using primers that contained the following sequences (Fw 5′-AGACCGGCAGATCTGATATCATCG ATG-3′, Rev 5′-TCGACGGTATCGATAAGCTTGATATCG-3′) to allow Gibson cloning into *Eco*RI-digested L4440.

**Phenotypic analysis and microscopy.** SCN was quantified in day-1 adults or late L4s. Animals were anaesthetised using 100 μM sodium azide and mounted on a 2% agarose pad. Seam cells were visualised using the *scm::GFP* marker (*wIs51*)[19] and the lateral side closer to the lens was counted on a Zeiss compound microscope (AxioScope A1) with 400x total magnification.

RILs were phenotyped over a period of one week as they were growing at different rates. 116 RILs were scored twice and 38 RILs were scored thrice to make sure that RILs included in the pools for sequencing were the ones with the most reproducible SCN between replicates. We pooled DNA of the two extreme groups (22 RILs in the low-SCN bulk and 24 RILs in the high-SCN bulk) for QTL mapping.

Single-molecule mRNA fluorescent in situ hybridisation was performed using a Cy5-labelled *elt-6* and *hsp-110* probes (Biomers). Z-stacks with 17–30 slices, each of 0.7 μm, were acquired with a ×100 oil immersion objective using an Andor iKon M 934 CCD camera system on a Nikon Ti Eclipse epifluorescence microscope using NIS elements (AR). Region of interests were drawn manually around seam cells visualised using the *scm::GFP* marker and MATLAB was used to quantify mRNA molecules[12]. Oligos for smFISH probes are shown as Supplementary Data 3 together with all other oligos used in this study.

Confocal images were obtained on a Leica SP5 with the LAS AF software using an Argon 488 nm laser and analysed with Fiji. A region of interest was drawn from the vulva to the tail containing intestinal, vulval and seam cells expressing the POPHHOP marker. The corrected total cell fluorescence (CTCF) was calculated as follows: CTCF = integrated density − (area of ROI × mean fluorescence of background readings)

**Quantitative genetics.** RILs were generated by crossing hermaphrodites from strain MBA256 (CB4856) and males from strain MBA231 (CB4856). F1 males from the cross were crossed to hermaphrodites from strain MBA290 (N2) containing the *egl-18(ga97)* mutation. Multiple F1 hermaphrodites, which were egg-laying defective and carried the transgene *vtIs1[dat-1::gfp] wIs51[scm::GFP]* V were picked and allowed to self as cross progeny. 117 F2s that were egg-laying and

carried only *wIs51[scm::GFP]* V transgene were picked onto single NGM plates and allowed to self. One hermaphrodite was transferred to a new plate for 10–14 generations to establish RILs. RIL-17 did not propagate during the selfing process so 116 RILs were produced in total.

We used a bulk segregant analysis approach to discover quantitative trait loci as previously done[34]. DNA was extracted from the low and high bulks using a Gentra Puregene Kit (Qiagen). Whole-genome sequencing was performed in an Illumina® Hiseq platform (read length = $2 \times 125$), aiming to obtain at least 10 million read pairs per sample (25× coverage). The whole-genome sequencing data were analysed through the CloudMap Hawaiian variant mapping pipeline on a local galaxy server maintained in the laboratory[65]. The genotypes for low-SCN and high-SCN was extracted into a single vcf file from the whole-genome data using vcf combine on galaxy. The combined vcf was converted to tabular format using GATK tools on a bash terminal. Frequency of CB4856-like variation at each SNP position was calculated as the ratio of read counts with CB4856 SNP divided by the total number of reads. For the low-SCN and high-SCN bulk, the SNP positions where the genome quality (as determined by GQ scores < 40) for the sequencing was low were discarded using the QTLseqr package in R[66]. SNPs where the total read depth was lower than 22 in the low-SCN bulk and lower than 24 in the high-SCN bulk were discarded[34]. We used log-odds ratio to evaluate if the deviations observed in SNP frequencies between the low-SCN and high-SCN bulk were statistically different to what is expected from a null distribution[34]. We calculated observed log-odds ratio using the following formula: log-odds ratio $= \log_{10}(\frac{h}{n_h - h} / \frac{l}{n_l - l})$, where $l$ and $h$ are mean SNP frequencies of CB4856 in a 300 kb window with no overlap multiplied by $n_l$ and $n_h$, $n_l$ and $n_h$ are number of RIL lines pooled for low-SCN and high-SCN bulks, respectively. The log-odds ratios under the null hypothesis were calculated where SNP frequencies for both the bulks were generated by 1 million simulated Bernoulli trails with $p = q = 0.5$. From these log-odds ratios, we found the two-tailed genome-wide threshold at a significance level ($\alpha = 0.05$). The observed log-odds ratio was plotted against the genome location and was compared with the genome-wide thresholds. The presence of a QTL was inferred when the log-odds ratio exceeded the threshold.

**CRISPR-Cas9-mediated genome editing**. *hsp-110* and *sor-1* genetic variants in the N2 background were generated via injection of CAS9 ribonucleoprotein complexes[67] at the following final concentrations in 10 μl: tracRNA (0.75 nmol—IDT), custom crRNAs (*hsp-110* target sequence–AAAGTCAATGGAGCACGACC and AACCTTCTGATTACCACCAT), (*sor-1*—TAGTCAACGCCCACCAAGCA, 20 μM, IDT) and Cas9 (1 μg/μl—IDT). The co-injection markers *myo-2::RFP* at 5 ng/μl and *rol-6(su1006)* at 40 ng/μl were used to select transgenic animals, and repair templates (6 μM) introducing SmaI sites in the case of *hsp-110* and ClaI in the case of *sor-1* together with the desired CB4856 SNP changes. F1 animals were screened for the co-injection markers, allowed to lay progeny and screened using restriction digests for the introduced sites. Positive lines were Sanger-sequenced to identify lines homozygous for SNP insertions. The co-CRISPR strategy[68] was used to edit *bro-1*. An sgRNA targeting the following sequence (5′-AATCAATA-TACCTGTCAAGT) was cloned into pU6::unc-119 sgRNA vector by replacing the *unc-119* sgRNA using PCR amplification and *Eco*RI/*Hin*dIII ligation[69]. The recovered *icb45* allele is an in-frame deletion of 9 bp (GGAATCAATATA- - - - - - - - -TTGGAATGGT) within the second exon of *bro-1*.

**Reporting summary**. Further information on research design is available in the Nature Research Reporting Summary linked to this article.

## Data availability
WGS data have been submitted to the Sequence Read Archive (SRA accession number PRJNA707208). Source data are provided with this paper.

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

## Acknowledgements

We thank Mingke Pan for technical help with genotyping and Lise Frézal, Marie-Anne Félix for discussions. Some *C. elegans* strains were provided by the CGC, which is funded by NIH Office of Research Infrastructure Programs (P40 OD010440). This work was funded by the European Research Council (ROBUSTNET-639485).

## Author contributions

S.K. performed the majority of experiments and data analysis. M.H and D.K. contributed to the generation of transgenic strains and phenotyping. M.B. supervised the work. All authors contributed to drafting and revising the manuscript.

## Competing interests

The authors declare no competing interests.
