## [Peer Review File · Nature Communications]

REVIEWER COMMENTS

Reviewer #1 (Remarks to the Author):

This study from the Barkoulas lab addresses how cryptic genetic variation, which normally has little effect on the phenotype, creates variable expressivity in the stem cell division phenotype associated with a specific mutation. The authors make use of independent isolates of the nematode *C. elegans*, specifically the standard laboratory strain from Bristol (N2) and a well-characterized isolate from Hawaii (CB4648). They notice that a null mutation of the EGL-18 transcription factor is better tolerated in the Hawaiian background, which they attribute to robustness in atypical Wnt signaling that acts upstream to control this transcription factor. Through the generation and characterization of a large number of recombinant inbred lines, and near isogenic lines, they manage to assign multiple quantitative trait loci (QTLs) that affect the expressivity of the *egl-18* mutant phenotype. For two of these loci, candidate genetic modifiers are identified (*dsh-2* encoding a Wnt pathway component and *egl-27* a chromatin regulator). For a third QTL, the responsible variant is identified to be a 3 nucleotide deletion in *hsp-110*, which removes a single conserved amino acid from the substrate binding site in a heat shock chaperone. Cas9/CRISPR-mediated introduction of this variant into the N2 background significantly increases the tolerance for the *egl-18* null mutation, thereby demonstrating that this is one of the contributing factors.

This is a very attractive study, the experimental work has been a substantial tour de force, the data are well represented and are technically solid, the conclusions are well supported by the results, and the entire manuscript is very well written. Although ready for publication as it is, I find it difficult to judge the impact to the field.

To summarize some considerations:

The study addresses a very important and poorly understood topic in biomedical research. Understanding the often complex relationship between genotype and phenotype is critical for comprehending the often variable expressivity of human diseases and developmental abnormalities, as well as the variable response to treatment.

The concept of cryptic genetic variation is not new but certainly temporal, and molecular insight in the underlying mechanisms is very much desired. In this regard, the current study only goes half-way: a single amino-acid deletion in HSP-110 is identified as a phenotype modifying natural variation. The effect of this variation is limited, although highly significant, but the other candidates are not followed up to possibly obtain a more complete switch. In addition, it remains unclear why this D474 deletion makes the Hawaiian strain more tolerant for the *egl-18(ga97)* mutation. One might expect the D474-deleted HSP-110 chaperone to be less functional and, simply thinking, it would appear more logical that this results in greater sensitivity to disruptions, instead of a reduced sensitivity. Obviously, the situation is more complex, but it remains somewhat unsatisfying that a deeper understanding of the molecular mechanisms behind the phenotypic variation is not obtained. Nevertheless, a major strength of the study is that at least one, and possible multiple molecular changes in the genetic background have been

identified that influence a specific phenotypic outcome.

The study also succeeds in demonstrating the complexity of natural genetic variation in phenotypic variation. For this aspect, the authors should probably give more credit to some earlier work published by others, in particular studies from the Kammenga lab with N2-Hawaiian inbred lines (e.g. Snoek et al 2017 Contribution of trans regulatory eQTL to cryptic genetic variation in *C. elegans* DOI: 10.1186/s12864-017-3899-8).

For relevance, it would have been nice if some effort was made to confirm that the *egl-18(ga97)* allele is a true null mutation, and that the genetic modifiers are not allele specific. Possibly the simplest experiment would be to cross the *hsp-110* mutation created in N2 into another *egl-18(null)* mutant background?

Minor point:

line 300: add "(NIL)" behind "near inbred lines" to explain the abbreviation used in later sentences.

Reviewer #2 (Remarks to the Author):

Summary: This manuscript outlines the genetic dissection of cryptic genetic variation in a key *C. elegans* developmental trait. The authors characterize the differences in "expressivity" caused by a null mutation in the seam cell fate-determining gene *egl-18* between the diverged N2 and CB4856 strains (RILs from these strains). Using a bulk segregant approach followed by the construction of near-isogenic lines, the authors identified three key QTL that independently decrease the expressivity of the *egl-18* phenotype by differentially potentiating or propagating Wnt signaling during development. The authors also functionally validated a derived single nucleotide variant in the CB4856 allele of *hsp-110*, which underlies one of these QTL, using CRISPR-mediated genome editing. This work will undoubtedly contribute to a growing knowledge base of cryptic genetic variation in *C. elegans*, moving from a complex phenotype to a single causal variant in one case. Although many of the results are described rigorously and accurately, we identified a few major and minor revisions that would strengthen or clarify the generalizability of the results.

Major Concerns:

- (1) The first major concern is one of semantics. Expressivity and penetrance refer to phenotypes not mutations. Penetrance refers to the fraction of mutants in a population that express the mutant phenotype, and expressivity refers to the level of mutant phenotype observed in the mutants. Figure 1B is an easy illustration of how both penetrance and (mostly) expressivity are altered. I suggest that the authors edit their title and much of the manuscript to reflect the proper use of the term. Just like the abuse of recessive and dominant has grown over the years, all of these terms should strictly be applied to phenotype or specific traits.
- (2) It is puzzling why the authors took all of the time to construct 116 RILs with their mutation of interest

and a reporter, genotype the RILs, and then phenotype them for seam cell defects but then never do a proper linkage mapping experiment. The authors should do this analysis along with calculating heritability to help the readers to understand more about this trait variance.

(3) The “bulk segregant approach” (BSA) is much less powerful than linkage mapping in this context and previous reports. Given the low level of recombination in *C. elegans*, the mapping resolution of detected QTL suffers. Some detected QTL are 8 Mb! Additionally, BSA does not give effect size of detected QTL, another disadvantage. More importantly, a clear explanation of why so few RILs were retained in the low (N2-like) bulk was not given, as the figure displays 22 RILs while Figure S2 displays 10. The claim that “RILs in the low bulk carrying large CB4856 fragments on chromosomes III and X also carried a CB4856 fragment on chromosome V” (287-292) doesn’t appear to be supported by the data. I only see 10 RILs present in the low bulk in this figure, and RIL-43 appears to be the only instance of CB4856 haplotypes shared across the three chromosomes in question. Extending on the above point, assigning genetic modes of action (additivity/epistasis) based on these data seems dubious. There is strong evidence for an

additive relationship among QTL in the NIL phenotype data, but the statistical relationship between these haplotypes and the phenotype has not been directly established across the population. The authors should clarify and address all of these points.

(4) Based on what is written in the methods (576-585), I’m not convinced that the significance threshold is correctly calculated. In Frezal et al. 2018, the null distribution of bulk allele frequencies was constructed by one million Bernoulli draws from the allele frequencies present in the two bulk pools, which seems correct and more analogous to permutation-based thresholding in traditional linkage analysis. Here, the threshold is constructed from Bernoulli trials under the assumption of exactly equal allele frequencies in both bulks. Given that 1) the bulks have unequal sample sizes and, therefore, certainly greater sampling error and 2) areas of the genome unrelated to the phenotype of interest have distorted ratios (peel/zeel and the *egl-18* allele), I feel an ideal approach would be to simulate these Bernoulli trials from an empirical sampling of allele frequencies present in the low and high bulk RILs.

(5) Early on in the results, the claim is made that “differential expressivity of the *egl-18(ga97)* mutation between N2 and CB4856 is independent of changes in *elt-6* expression” (195-197). The authors, however, correctly note earlier that *elt-6* could only modulate *egl-18* expressivity due to expression changes in trans because the two loci are linked (192). The following result that *elt-6* expression does not differ in the N2 and CB4856 *egl-18* mutants is not surprising, but it does not rule out cis-e QTLs of the *elt-6* locus that segregate between the strains (just not in this experiment). Changes in cis in this redundant gene are 1) undetectable in this experiment and 2) a more likely source of variation than changes in trans. I am almost 100% convinced of the proposed mechanism - CB4856 is differentially “primed” to promote Wnt signaling and seam cell fate canalization - but this important limitation is not hedged properly in my opinion.

Minor Concerns:

(1) The authors highlight in the introduction that seam maintenance can serve as a simplified model of stem cell patterning (82-83). The contribution of this study to the model is briefly mentioned in the discussion (450 – 451). In order to impress upon readers the impact this study has on the stem cell model further elaboration of this point is key and should contain information about the limitations of this simplified seam cell model.

(2) The colors used in the figures do not match Rockman and other published works. N2 should be orange, and CB4856 should be blue.

Specific comments:

Introduction

53, and throughout: phenotypic traits?

55: aims to exploit*

65: I would like to see a clearer connection between the subtext (cryptic variation contributes to complex traits in the context of medicine) and how modifiers of this specific large-effect mutation is important. Consider switching P3 and P2 to create a clearer connection from medical genetics (P1) → seam cell model for cancer biology (P2) → natural variation in mutational expressivity matters (P3 + P4)

73: phenotypes and* nath-10

75-78: Consider rephrasing for clarity.

82: "We study here" doesn't really work for me.

Results

153: Difficult to evaluate bro-1 or eff-1 result from provided supplemental figure - alleles being compared on x-axis discordant from gene names. Consider distinguishing gene, mutation, and strain.

155-156: "investigate in more detail" is not a proper predicate.

171-175: Consider a clearer outline of these hypotheses in two sentences; multiple phrases confused me.

186-187: its* paralogue

190: in of in trans s hould not be italicized, same goes for in cis

Figure 3: No idea how to interpret the -LOD score. Why were only 10 RILs present in the low bulk?

Figure 3A: Why not use Tukey box plots?

321: Given that WT strains were employed in the previous section, it would be helpful to remind the reader here that pop-1 RNAi was conducted in egl18(ga97) NILs, not WT NILs (because the variation is cryptic).

350: This should maybe be softened to "candidates" considering no follow up with CRISPR as with Chr III.

Discussion:

391: Double period

423-424: What is meant by "genetic disease"? Mendelian genetic disorders in humans/broadly? Genetic risk factors genome-wide? How generalizable?

439-441: Like above, the dissection of compensation by elt-6 was someone limited in scope since the allele (like egl18(ga97)) did not segregate in the mapping population and NILs and was WT in all comparisons. This, importantly, does not invalidate the modifier effects of other QTL, but it also does not completely rule out the possibility of cis-acting elt-6 expression.

Figure 6: A more detailed pathway would emphasize the hypothesis in the context of cryptic variation more effectively. A couple of suggestions:

- pathway diagrams specific to CB4856 and N2, with and without the egl-18 LOF allele - where other QTL work in the model besides Chr III locus (which seems to have the clearest effect from pop-1 RNAi and CRISPR)

Materials and Methods:

RILs - Proper strain names should be assigned by the lab

498: Do egl-18 animals not respond in bleach? Or vulval phenotype is too severe? 506-507: Is this a thing?

500, 548: Something off here. 1) was JU2007 in this study? 2) N2 was surely crossed to CB4856 but proper lab codes are needed in every case.

563: genotypes*

614: difference in seam cell abundance*?

615: at the L1 stage*?

We thank the reviewers for providing thoughtful comments and suggestions in these trying times. Here is our point-by-point response to all comments:

Reviewer #1 (Remarks to the Author):

This study from the Barkoulas lab addresses how cryptic genetic variation, which normally has little effect on the phenotype, creates variable expressivity in the stem cell division phenotype associated with a specific mutation. The authors make use of independent isolates of the nematode *C. elegans*, specifically the standard laboratory strain from Bristol (N2) and a well-characterized isolate from Hawaii (CB4648). They notice that a null mutation of the EGL-18 transcription factor is better tolerated in the Hawaiian background, which they attribute to robustness in atypical Wnt signaling that acts upstream to control this transcription factor. Through the generation and characterization of a large number of recombinant inbred lines, and near isogenic lines, they manage to assign multiple quantitative trait loci (QTLs) that affect the expressivity of the *egl-18* mutant phenotype. For two of these loci, candidate genetic modifiers are identified (*dsh-2* encoding a Wnt pathway component and *egl-27* a chromatin regulator). For a third QTL, the responsible variant is identified to be a 3 nucleotide deletion in *hsp-110*, which removes a single conserved amino acid from the substrate binding site in a heat shock chaperone. Cas9/CRISPR-mediated introduction of this variant into the N2 background significantly increases the tolerance for the *egl-18* null mutation, thereby demonstrating that this is one of the contributing factors. This is a very attractive study, the experimental work has been a substantial tour de force, the data are well represented and are technically solid, the conclusions are well supported by the results, and the entire manuscript is very well written. Although ready for publication as it is, I find it difficult to judge the impact to the field.

Thank you for your comments. Heat shock proteins have been previously proposed to buffer phenotypic variation and modify the genotype-to-phenotype relationship in many organisms. Our work moves the field forward because it provides an example where natural variation in a conserved heat shock protein modulates phenotype expressivity. In addition to this, our model introduces the exciting possibility that heat shock proteins may act as genetic modifiers of Wnt signalling.

To summarize some considerations:

The study addresses a very important and poorly understood topic in biomedical research. Understanding the often complex relationship between genotype and phenotype is critical for comprehending the often variable expressivity of human diseases and developmental abnormalities, as well as the variable response to treatment. The concept of cryptic genetic variation is not new but certainly temporal, and molecular insight in the underlying mechanisms is very much desired. In this regard, the current study only goes half-way: a single amino-acid deletion in HSP-110 is identified as a phenotype modifying natural variation. The effect of this variation is limited, although highly significant, but the other candidates are not followed up to possibly obtain a more complete switch. In addition, it remains unclear why this D474 deletion makes the Hawaiian strain more tolerant for the *egl-18(ga97)* mutation. One might expect the D474-deleted HSP-110 chaperone to be less functional and, simply thinking, it would appear more logical that this results in greater sensitivity to disruptions, instead of a reduced sensitivity. Obviously, the situation is more complex, but it remains somewhat unsatisfying that a deeper understanding of the molecular mechanisms behind the phenotypic variation is not obtained. Nevertheless, a major strength of the study is that at least one, and possible multiple molecular changes in the genetic background have been identified that influence a specific phenotypic outcome.

Although cryptic genetic variation has been revealed in other contexts during *C. elegans* development, in most cases its molecular basis remains poorly characterised. Our results are consistent with a model that natural variation in *hsp-110* modifies mutation expressivity via

potentiating the effect of the conserved Wnt signalling pathway. In the revised version, we have strengthened the link between HSP-110 and Wnt. We demonstrate increased sensitivity to *pop-1* RNAi in a strain carrying the CB4856 *hsp-110* allele. We also report a synthetic interaction between the β -catenin *bar-1* and *hsp-110*, which suggests that BAR-1 and HSP-110 act in parallel to promote Wnt signalling and seam cell fate. These new results further strengthen our model, which we discuss on page 18/19. Understanding the exact mechanistic link between HSP-110 and Wnt based on the molecular consequences of the D474 deletion is certainly very interesting and this will require biochemical work as part of our future work.

The study also succeeds in demonstrating the complexity of natural genetic variation in phenotypic variation. For this aspect, the authors should probably give more credit to some earlier work published by others, in particular studies from the Kammenga lab with N2-Hawaiian inbred lines (e.g. Snoek et al 2017 Contribution of trans regulatory eQTL to cryptic genetic variation in *C. elegans* DOI: 10.1186/s12864-017-3899-8).

We now cite this study in the context of cryptic genetic variation.

For relevance, it would have been nice if some effort was made to confirm that the *egl-18(ga97)* allele is a true null mutation, and that the genetic modifiers are not allele specific. Possibly the simplest experiment would be to cross the *hsp-110* mutation created in N2 into another *egl-18*(null) mutant background?

Since no deletion of the *egl-18* locus exists, it is unclear whether *ga97* represents the null phenotype for *egl-18* so to be cautious about this we call it a putative null in the manuscript. When we first designed the genetic introgressions, we reasoned that *ga97* would be one of the strongest loss-of-function alleles available based on the frequency of the Egl and abnormal vulva phenotype in previous reports (PMID 11063687). To test this hypothesis, we now crossed the seam cell marker into new *egl-18* alleles from the stock centre (*n162* and *n474*). We found that *egl-18(ga97)* is indeed the strongest allele available based on seam cell phenotype (Fig. S6b). The difference in seam cell number expressivity between N2 and CB4856 relies on strong loss of *egl-18* function, for example we do not see it using the *egl-18(ok290)* deletion (Fig. S1B) or upon *egl-18* RNAi, so we wouldn't expect to see a difference when using milder than *ga97* *egl-18* alleles. The fact that there is allele-specificity does not decrease the importance of the findings since an *egl-18* mutation is not naturally present in CB4856 – it just provided a tool for a difference in phenotype expressivity to manifest. Importantly, we also found that *hsp-110* interacts in the same manner with other *egl-18* alleles, with *hsp-110* RNAi amplifying the decrease in seam cell number observed upon loss of *egl-18* function (Fig. S6b).

Minor point:

line 300: add "(NIL)" behind "near inbred lines" to explain the abbreviation used in later sentences.

OK

Reviewer #2:

Summary: This manuscript outlines the genetic dissection of cryptic genetic variation in a key *C. elegans* developmental trait. The authors characterize the differences in "expressivity" caused by a null mutation in the seam cell fate-determining gene *egl-18* between the diverged N2 and CB4856 strains (RILs from these strains). Using a bulk segregant approach followed by the construction of near-isogenic lines, the authors identified three key QTL that independently decrease the expressivity of the *egl-18* phenotype by differentially potentiating or propagating Wnt signaling during development. The authors also functionally validated a derived single nucleotide variant in the CB4856 allele of *hsp-110*, which underlies one of these QTL, using CRISPR-mediated genome editing. This work will undoubtedly contribute to a growing knowledge base of cryptic genetic variation in *C. elegans*, moving from a complex phenotype to a single causal variant in one case. Although many of the results are described

rigorously and accurately, we identified a few major and minor revisions that would strengthen or clarify the generalizability of the results.

Thank you for your comments.

Major Concerns:

(1) The first major concern is one of semantics. Expressivity and penetrance refer to phenotypes not mutations. Penetrance refers to the fraction of mutants in a population that express the mutant phenotype, and expressivity refers to the level of mutant phenotype observed in the mutants. Figure 1B is an easy illustration of how both penetrance and (mostly) expressivity are altered. I suggest that the authors edit their title and much of the manuscript to reflect the proper use of the term. Just like the abuse of recessive and dominant has grown over the years, all of these terms should strictly be applied to phenotype or specific traits.

Thank you for this comment. We have now edited the title and all of the manuscript to reflect the formal definition of these terms based on phenotype and not mutation. We have also added a comment at the beginning of the results section to clarify that we treat the difference between isolates as a difference in phenotype expressivity although we cannot rule out that there is a difference in penetrance too. We think this is a fair assumption because *egl-18* mutant animals showing the WT number of 16 cells per lateral side are infrequent in the population. Also, even when 16 seam cells are present per lateral side, it is still possible that these are generated through defective patterning events so, strictly speaking, the lineage may not be truly wild-type.

(2) It is puzzling why the authors took all of the time to construct 116 RILs with their mutation of interest and a reporter, genotype the RILs, and then phenotype them for seam cell defects but then never do a proper linkage mapping experiment. The authors should do this analysis along with calculating heritability to help the readers to understand more about this trait variance.

We have addressed this comment as follows. We now present seam cell number counts in *egl-18* mutants in N2 and CB4856 from multiple independent experiments to show that the difference in phenotype expressivity is very reproducible and this is what inspired us to pursue this project (Fig. S1a). We also plotted correlation in independent phenotypic scorings of the RILs to highlight the reproducibility despite the arguably subtle phenotype (Fig. S3a). Since there is no phenotypic difference between the wild-type strains, we calculated heritability based on the replicated measures in our RILs and indeed there is a substantial heritable genetic component ($H^2=70.43\%$), as expected since we were able to identify QTLs. Just to clarify to the reviewer that we did not sequence each RIL independently.

(3) The “bulk segregant approach” (BSA) is much less powerful than linkage mapping in this context and previous reports. Given the low level of recombination in *C. elegans*, the mapping resolution of detected QTL suffers. Some detected QTL are 8 Mb! Additionally, BSA does not give effect size of detected QTL, another disadvantage. More importantly, a clear explanation of why so few RILs were retained in the low (N2-like) bulk was not given, as the figure displays 22 RILs while Figure S2 displays 10. The claim that “RILs in the low bulk carrying large CB4856 fragments on chromosomes III and X also carried a CB4856 fragment on chromosome V” (287-292) doesn’t appear to be supported by the data. I only see 10 RILs present in the low bulk in this figure, and RIL-43 appears to be the only instance of CB4856 haplotypes shared across the three chromosomes in question. Extending on the above point, assigning genetic modes of action (additivity/epistasis) based on these data seems dubious. There is strong evidence for an additive relationship among QTL in the NIL phenotype data, but the statistical relationship between these haplotypes and the phenotype has not been directly established across the population. The authors should clarify and address all of these points.

With regard to the number of RILs, there were 22 RILs included in the low bulk and 24 RILs in the high SCN bulk. Out of these, 10 and 24 lines respectively were used for depooling and

genotyping. In the revised version, we have removed the description of initial observations from depooling since this was not adding much to the manuscript. We have also edited the text to be strict in our discussion and added stats in Fig. 5a to show comparisons between NILs when significant.

(4) Based on what is written in the methods (576-585), I'm not convinced that the significance threshold is correctly calculated. In Frezal et al. 2018, the null distribution of bulk allele frequencies was constructed by one million Bernoulli draws from the allele frequencies present in the two bulk pools, which seems correct and more analogous to permutation-based thresholding in traditional linkage analysis. Here, the threshold is constructed from Bernoulli trials under the assumption of exactly equal allele frequencies in both bulks. Given that 1) the bulks have unequal sample sizes and, therefore, certainly greater sampling error and 2) areas of the genome unrelated to the phenotype of interest have distorted ratios (peel/zeel and the egl-18 allele), I feel an ideal approach would be to simulate these Bernoulli trials from an empirical sampling of allele frequencies present in the low and high bulk RILs. What we have done is exactly the same with what is presented in Frezal et al. 2018 (in fact we used the very same R code). The two bulks have almost the same size (22 RILs in the low bulk and 24 RILs in the high SCN bulk) which is also approximately the same with the design in Frezal et al. 2018 (22 vs 21 RILs). We have plotted again the graphs with threshold calculated by resampling the empirical data 10,000 times from combined SNP frequencies of low and high bulk and calculating the LOD score. As you can see in the graphs below, the results with regard to the identified regions of interest are very similar.

(5) Early on in the results, the claim is made that “differential expressivity of the egl-18(ga97) mutation between N2 and CB4856 is independent of changes in elt-6 expression” (195-197). The authors, however, correctly note earlier that elt-6 could only modulate egl-18 expressivity due to expression changes in trans because the two loci are linked (192). The following result

that *elt-6* expression does not differ in the N2 and CB4856 *egl-18* mutants is not surprising, but it does not rule out cis-e QTLs of the *elt-6* locus that segregate between the strains (just not in this experiment). Changes in cis in this redundant gene are 1) undetectable in this experiment and 2) a more likely source of variation than changes in trans. I am almost 100% convinced of the proposed mechanism - CB4856 is differentially “primed” to promote Wnt signaling and seam cell fate canalization - but this important limitation is not hedged properly in my opinion.

To address this point, we performed *egl-18* and *elt-6* mRNA quantifications by smFISH in wild-type N2 and CB4856. We found that there is no significant difference in the expression of these two transcription factors between the two isolates (Fig. S2a,b). This argues against cis-changes on the *elt-6* locus affecting its levels of expression between N2 and CB4856. Because of the close linkage between *egl-18* and *elt-6*, we cannot formally rule out that the less severe outcome of the *egl-18(ga97)* mutation in CB4856 is only observed when the *elt-6* locus from N2 is present and we now discuss this limitation on page 18...“Although we did find an increase.....potentiate the Wnt signalling pathway to promote the seam cell fate”.

Minor Concerns:

(1) The authors highlight in the introduction that seam maintenance can serve as a simplified model of stem cell patterning (82-83). The contribution of this study to the model is briefly mentioned in the discussion (450 – 451). In order to impress upon readers the impact this study has on the stem cell model further elaboration of this point is key and should contain information about the limitations of this simplified seam cell model.

We have added a comment and rephrased to highlight the potential value of the model.

(2) The colors used in the figures do not match Rockman and other published works. N2 should be orange, and CB4856 should be blue.

We used a colour-blind-friendly scheme that is consistent with our previous work (PMID: 31988193) and is also consistent throughout the paper. There are many published papers that do not strictly follow this “convention” (e.g. PMID: 28179390, PMID: 26186192 etc.) so we were not aware of this. We have decided to leave the colour scheme as it is to avoid remaking all figures again, but we will keep this suggestion in mind for the future.

Specific comments:

Introduction

53, and throughout: phenotypic traits?

OK

55: aims to exploit*

OK

65: I would like to see a clearer connection between the subtext (cryptic variation contributes to complex traits in the context of medicine) and how modifiers of this specific large-effect mutation is important. Consider switching P3 and P2 to create a clearer connection from medical genetics (P1) → seam cell model for cancer biology (P2) → natural variation in mutational expressivity matters (P3 + P4)

OK, we have improved the flow by changing the order of some paragraphs.

73: phenotypes and* *nath-10*

OK

75-78: Consider rephrasing for clarity.

We rephrased to add clarity

82: “We study here” doesn’t really work for me.

OK

Results

153: Difficult to evaluate *bro-1* or *eff-1* result from provided supplemental figure - alleles

being compared on x-axis discordant from gene names. Consider distinguishing gene, mutation, and strain.

We added gene names to help the reader.

155-156: "investigate in more detail" is not a proper predicate.

OK

171-175: Consider a clearer outline of these hypotheses in two sentences; multiple phrases confused me.

OK, corrected

186-187: its* paralogue

OK

190: in of in trans s hould not be italicized, same goes for in cis

OK

Figure 3: No idea how to interpret the -LOD score. Why were only 10 RILs present in the low bulk?

Negative LOD scores are obtained when the low-SCN bulk has a higher proportion of CB4856 SNPs compared to the high-SCN bulk. This should be interpreted as CB4856 SNPs being associated with low SCN so potentially having a negative effect on the seam cell number phenotype. We have added a comment in the figure legend. With regard to the number of RILs in the low bulk, there are 22 RILs included for bulk segregant analysis (10 of which were used for depooling and genotyping).

Figure 3A: Why not use Tukey box plots?

Box and whisker plots show median, which is not the statistic being compared in the experiment. We show mean and confidence intervals, as well as all data points in all graphs, as per the guidelines for figure preparation for this journal.

321: Given that WT strains were employed in the previous section, it would be helpful to remind the reader here that *pop-1* RNAi was conducted in *egl18(ga97)* NILs, not WT NILs (because the variation is cryptic).

In fact, *pop-1* RNAi was performed in WT NILs (not in an *egl-18(ga97)* background) and we have rephrased to clarify. We found that *pop-1* RNAi is another way to sensitise the system.

350: This should maybe be softened to "candidates" considering no follow up with CRISPR as with Chr III.

OK

Discussion:

391: Double period

OK

423-424: What is meant by "genetic disease"? Mendelian genetic disorders in humans/broadly? Genetic risk factors genome-wide? How generalizable?

OK

439-441: Like above, the dissection of compensation by *elt-6* was someone limited in scope since the allele (like *egl18(ga97)*) did not segregate in the mapping population and NILs and was WT in all comparisons. This, importantly, does not invalidate the modifier effects of other QTL, but it also does not completely rule out the possibility of cis-acting *elt-6* expression.

(See also point 5 above) We have added a comment to clarify that *elt-6* is not segregating in this experiment. We have also added quantifications by smFISH showing that there is no difference in *elt-6* levels between wild-type N2 and CB4856 (Fig. S2B).

Figure 6: A more detailed pathway would emphasize the hypothesis in the context of cryptic variation more effectively. A couple of suggestions:

- pathway diagrams specific to CB4856 and N2, with and without the *egl-18* LOF allele - where other QTL work in the model besides Chr III locus (which seems to have the clearest effect from *pop-1* RNAi and CRISPR)

We have modified the summary figure to add clarity.

Materials and Methods:

Proper strain names should be assigned by the lab

We have done so and show MBA numbers for all NILs.

498: Do *egl-18* animals not respond in bleach? Or vulval phenotype is too severe? 506-507: Is this a thing?

Yes, it is possible to bleach *egl-18* animals, however these animals are a little more sensitive to bleaching so occasionally embryonic lethality can be observed. As crazy as it sounds, the most effective way to solve this problem was to cut open gravid *egl-18* mutant animals to release eggs!

500, 548: Something off here. 1) was JU2007 in this study? 2) N2 was surely crossed to CB4856 but proper lab codes are needed in every case.

Yes, JU2007 was used in Fig. S1. The MBA strain names listed follow the formal nomenclature.

563: genotypes*

OK

614: difference in seam cell abundance*?

OK

615: at the L1 stage*?

OK

Reviewer 3:

This is a very nice paper demonstrating that the requirement for the EGL-18 GATA factor in postembryonic development of the stem-cell-like epidermal “seam cells” in *C. elegans* varies between wild isolates. Specifically, the widely used Hawaiian strain shows lower sensitivity to removal of this transcription factor than does the standard laboratory N2 strain. The authors show that multiple QTLs underlie this variation, at least partially through potentiating Wnt signaling in the seam cell lineages. One of the causal QTLs is convincingly shown to be the result of a deletion in HSP-110 in the Hawaiian strain, supporting the view that heat-shock proteins can influence developmental variation. The paper is well-executed, the data are compelling, and the connection between HSP-110 and variation in seam cell development is of high interest to our understanding of developmental fidelity.

Thank you for your comments.

I recommend that the authors address the following points prior to publication.
1. Although the authors demonstrated the causal role of one of the QTLs, they did not perform any experiments with that allele and mutations in Wnt activity. This would be a straightforward experiment to perform and would enhance the claims in the paper. How might this QTL affect Wnt function?

To address this comment, we tested for possible interactions between HSP-110 and Wnt components. We found that the CB4856 *hsp-110* allele increases the sensitivity to reduction of *pop-1* by RNAi in comparison to N2 (Fig. 6G). We also found that *hsp-110* RNAi potentiates the loss of the beta-catenin *bar-1* (Fig. 6F), which is a very striking result since *bar-1* was not previously thought to play a role in seam cell development. We report these new findings in the revised Figure 6. Interestingly, related heat shock proteins have been described in other systems to act at the level of b-catenin activation (PMID: 25645927, PMID: 27279016). Taken together, these results strengthen our model that HSP-110 potentiates Wnt signalling. Understanding how HSP-110 mechanistically interacts with the Wnt pathway will require biochemical work as part of future work.

2. Line 148: this statement should be qualified. Although they show quantitatively similar results, it is possible that JU2007 and N2 have converged on quantitatively similar phenotypes by different genetic mechanisms.

We agree with the reviewer so we toned down the sentence.

3. Line 181: “These results suggest that *egl-18(ga97)* mutants show higher terminal seam cell

number in the CB4856 background compared to N2 due to higher cell fate retention across all cell lineages.” Cell fate “retention” is a confusing word to use here.

Rephrased to add clarity

4. Line 150/Fig. S1: it would be useful to describe the nature of the two alleles in the text (shown in Fig. S1).

Added this information in the text.

5. The authors should comment on the low variance in Hw in general with both alleles (Fig. S1A and address the variation in this variance. In addition, the authors should discuss why extra seam cells (up to 3) are (unexpectedly) seen in some cases.

The low variance in CB4856 carrying both *egl-18* alleles is somewhat expected based on the relationship between variance and mean for this phenotype. We have previously shown that the phenotypic variance is low around the wild-type mean of 16 and increases as seam cell number departs from the mean to any direction (increase or decrease, see PMID: 29108019). We added a comment about this in the legend of Fig. S1.

Cases of *egl-18* mutants displaying seam cell counts above 16 are rare but do indeed occur. Increase in seam cell counts upon loss of Wnt components has been reported in other contexts too, for example in *lin-17* loss of function mutants (PMID: 31988193). This may reflect compensation in the pathway, for example we report that *elt-6* levels increase by smFISH in *egl-18* mutants compared to wild-type. Putative compensation has been proposed for other Wnt mutants too (PMID: 24819947). Alternatively, this rare mild increase can also reflect low penetrant changes in seam cell polarity and cell division patterns.

6. Line 164. When EGL-18 is removed there is no effect of SCN in L1s, but the authors did not show results for wild-type CB486. They should confirm that WT Hw doesn't have different SCN in L1s (Fig. 1C).

Seam cell number is similar between CB4856 and N2 wild-type strains at the L1 stage too. We added the data in Fig. 1C for completion.

7. Lines 186-87 loss of it paralogue *egl-18* ... loss of its paralogue

Corrected.

8. Line 198: it would be appropriate to mention here that natural variation in Wnt requirement has been observed (e.g., Torres Cleuren et al., 2019).

We added this citation at this point to make a smoother transition to Wnt.

9. Line 210 refers to wrong figure: this should be 1G not S1E.

Corrected.

10. Line 391 there is an extra “.”

Corrected.

11. Lines 350-51 and lines 475-477: The authors found that RNAi of *egl-27* and *dsh-2*, which are known to be involved in seam cell development, resulted in decreased SCN in *egl-18(-)* and suggest that variation in these two genes may underly this effect. If this is the case, then there would be expected to be sequence variants in these genes that might explain or support this claim. Are there any interesting sequence variants in the sequences between the two strains that might support such a role?

Yes, there are multiple variants in *dsh-2* (exons 3 and 8) and *egl-27* (exons 11 and 12) in CB4856. These variants are presented in Data S3. We also clarified in the manuscript that we picked candidates to knock-down by RNAi among those that show natural variation between N2 and CB4856.

12. Line 421 The action of hsp90 as a developmental capacitor is somewhat controversial, curiously as has been noted in the literature by the senior author (Félix and Barkoulas, 2015), and this statement should be qualified accordingly.

Agreed, so we removed the term capacitor.

13. Line 448: Although Torres Cleuren et al. showed that the requirement for Wnt signaling varies in gut development, they actually showed that there is no variation in the requirement for POP-1 action across many strains, as should be corrected here.

We corrected this point.

14. Line 510: at what stage are they feeding RNAi?

We added a comment in the materials and methods to clarify this point.

REVIEWERS' COMMENTS

Reviewer #1 (Remarks to the Author):

In this revised manuscript from the Barkoulas lab, my previous comments have been more or less addressed. I was enthusiastic about this study the first time around, and consider it further improved.

Reviewer #2 (Remarks to the Author):

The authors have addressed our concerns.

Reviewer #3 (Remarks to the Author):

The authors have revised the manuscript to address reviewers' concerns effectively. I support publication of the revision.